# Noninvasive spinal stimulation safely enables upright posture in children with spinal cord injury

Anastasia Keller[1,2], Goutam Singh [1,2], Joel H. Sommerfeld [1,2], Molly King [1,2], Parth Parikh [2], Beatrice Ugiliweneza[1,2,3], Jessica D'Amico[1,2], Yury Gerasimenko[2,4,5] & Andrea L. Behrman [1,2,6✉]

In children with spinal cord injury (SCI), scoliosis due to trunk muscle paralysis frequently requires surgical treatment. Transcutaneous spinal stimulation enables trunk stability in adults with SCI and may pose a non-invasive preventative therapeutic alternative. This non-randomized, non-blinded pilot clinical trial (NCT03975634) determined the safety and efficacy of transcutaneous spinal stimulation to enable upright sitting posture in 8 children with trunk control impairment due to acquired SCI using within-subject repeated measures study design. Primary safety and efficacy outcomes (pain, hemodynamics stability, skin irritation, trunk kinematics) and secondary outcomes (center of pressure displacement, compliance rate) were assessed within the pre-specified endpoints. One participant did not complete the study due to pain with stimulation on the first day. One episode of autonomic dysreflexia during stimulation was recorded. Following hemodynamic normalization, the participant completed the study. Overall, spinal stimulation was well-tolerated and enabled upright sitting posture in 7 out of the 8 participants.

[1] Department of Neurological Surgery, University of Louisville, Louisville, KY, USA. [2] Kentucky Spinal Cord Injury Research Center, University of Louisville, Louisville, KY, USA. [3] Department of Health Management and Systems Science, University of Louisville, Louisville, KY, USA. [4] Department of Physiology, University of Louisville, Louisville, KY, USA. [5] Pavlov Institute of Physiology, St. Petersburg, Russia. [6] Kosair Charities Center for Pediatric NeuroRecovery, University of Louisville, Louisville, KY, USA. ✉email: andrea.behrman@louisville.edu

While pediatric spinal cord injury (SCI) is rare (less than 5% of all SCI cases), the lifelong impact of paralysis, chronic physiological dysfunction, and an imposed sedentary state make healthcare utilization very high in this population[1]. Current therapeutic interventions largely aim to compensate for paralysis assuming its permanence as a result of irreversible damage to the central nervous system[1–4]. These interventions fail to prevent the unique secondary health complications following pediatric-onset SCI. In particular, trunk muscle paralysis is associated with a 100% incidence rate of neuromuscular scoliosis in children injured under the age of 10[5–7]. Scoliosis onset and progression accelerates functional decline and can cause severe respiratory compromise in children with 67% eventually requiring invasive surgical correction of the scoliotic curve[8,9].

The discovery of the "intelligent" spinal cord that contains complex neuronal networks capable of generating rhythmic and coordinated motor patterns[10,11], known as the central pattern generator (CPG) for locomotion, has set forth a major paradigm shift in our expectation of what is possible in terms of recovery after even the most severe SCI[12,13]. Studies have demonstrated, first, that after SCI, the CPG can be accessed, reactivated, and retrained via sensory feedback arising from the muscles and joints during activity-based locomotor training[14–19]. Second, epidural and transcutaneous stimulation of the spinal cord below the level of the lesion can augment the neuromuscular capacity for voluntary movement, standing, and stepping in individuals with chronic motor complete SCI[13,20–22].

The majority of rehabilitation research efforts involving human subjects have focused on the adult population with SCI. The field of pediatric rehabilitation lacks high-quality empirical investigations that determine optimal neurotherapeutic interventions for children with SCI[23,24]. Meanwhile, children with SCI may significantly benefit from advanced therapeutic approaches and gain improvements in neuromuscular recovery due to inherent plasticity present in the stages of rapid growth and development[25,26].

Transcutaneous spinal stimulation (scTS) is a promising, non-invasive technology for neuromodulation that may be accessible and beneficial to children[27–29]. Rath et al.[30] demonstrated that scTS improves sitting posture and trunk control in adults with chronic SCI. In children with cerebral palsy, robotic-assisted locomotor training in combination with scTS resulted in greater improvement of locomotion as compared to locomotor training alone[31]. More recently, Baindurashvili et al.[32] implemented transcutaneous spinal stimulation as a part of a comprehensive rehabilitation regimen as early as 8 days post-SCI (AIS B) sustained by a 17-year-old male during roller ski training. After a year of intensive rehabilitation, the participant was able to stand unassisted and walk with a cane. Stimulation in these studies was generally reported as safe and well-tolerated by the participants,

with the exception of one incident of systolic blood pressure (BP) increase over 60 mmHg during stimulation reported in a study examining the effects of scTS on standing capacity in adults with SCI[33]. In the same study, a small skin breakdown occurred at the site of spinal stimulation in one of the participants following a training session due to a defect in the conduction layer of the stimulating electrode[33]. Otherwise, participants reported no pain or discomfort during scTS[33]. The evidence of scTS efficacy to facilitate neuromuscular function makes it an attractive neuro-modulation tool to facilitate recovery of intrinsic trunk control in children with SCI. Since children with SCI represent a vulnerable population, we first must clearly establish the safety and feasibility of scTS as a potential therapeutic approach.

In this work, we demonstrate the safety (absence of major adverse events in response to continuous spinal cord stimulation), feasibility (most children tolerate stimulation well), and efficacy of transcutaneous spinal cord stimulation to acutely potentiate upright sitting posture in children with SCI.

## Results

**Safety and feasibility of acute scTS in children with SCI.** The study compliance and attendance was 92% with 22/24 experiments attended. Seven out of 8 participants with SCI tolerated scTS across a wide range of stimulation intensities at T11, L1, and C5 spinal levels (Table 1)[34]. At low scTS intensity levels (20–37 mA) at T11, P21 reported feeling a "burning", needle prick-like sensation at the site of stimulation. Removal of the stimulation immediately resolved the participant's report of painful stimulation. The experiment was terminated, and P21 was excluded from days 2 and 3 of the assessments. Two out of seven remaining participants self-reported discomfort/pain with scTS at the C5 location in which case the scTS intensity was turned down/off. However, all participants reported 0 pain at the standardized time points of pain assessment within the experiments, including with scTS on at T11, L1, and C5 sites. One episode of autonomic dysreflexia (AD) was observed in participant P49 who presented with a rapid onset of facial redness during scTS testing at C5. P49's BP was measured and recorded at 134/82 mmHg with a heart rate (HR) of 58 beats per minute (bpm). There was a 27 mmHg increase in systolic BP with a concomitant decrease in HR by 40 bpm from baseline measurement (107/67 mmHg, HR 98 bpm). The stimulation was stopped and the participant was monitored carefully. Within 3 min the episode resolved, with a follow-up BP of 114/82 mmHg and an HR of 94 bpm. Facial flushing or goosebumps were observed in other participants during stimulation without changes in BP (Table 2). Overall, there were no significant changes in either systolic ($F_{2,12} = 1.98$, $p = 0.18$), diastolic ($F_{2,12} = 2.27$, $p = 0.1462$) BPs, or HR ($F_{2,12} = 0.2$, $p = 0.82$) between the designated assessment

**Table 1 Participant demographics and stimulation parameters.**

| ID | Age (years) | SCI level | SATCO score | scTS T11 (mA) | scTS L1 (mA) | scTS C5 (mA) |
|----|-------------|-----------|-------------|---------------|--------------|--------------|
| P49 | 3 | Cervical | 12 | 100 | 70 | 20 |
| P34 | 4 | Thoracic | 12 | 140 | 160 | 20 |
| P23 | 6 | Cervical | 11 | 150 | 150[a] | 60 |
| P14 | 9 | Cervical | 11 | 130 | 140 | 65 |
| P4 | 9 | Cervical | 11 | 130 | 120 | NA |
| P21[b] | 9 | Thoracic | 19 | 37 | 50 | NA |
| P1 | 13 | Thoracic | 9 | 165 | 165 | 35 |
| P5 | 14 | Thoracic | 8 | 170 | 160 | 25 |

NA not assessed, mA milliamps.
[a]Optimal stimulation frequency at 30 Hz.
[b]Participant was not able to tolerate higher intensities due to skin allodynia in the region of stimulation.

**Table 2 Incidence of goosebumps during stimulation.**

| ID | Session# | Goose bumps | Facial flushing | Location | scTs T11 (mA) | scTs L1 (mA) | scTs C5 (mA) | Hemodynamics measurements (SBP/DBP, HR) |
|---|---|---|---|---|---|---|---|---|
| P5 | 2 | Y | N | Missing note | 120 | | | 120/69, 110 |
| P4 | 2 | Y | N | LE bilaterally | 110 | 100 | | 110/82, 83 |
| P14 | 1 | Y | N | Right lower trunk | 100 | | | 87/53, 76 |
| P14 | 2 | Y | N | Trunk bilaterally | 60 | | | 95/56, 61 |
| P14 | 3 | Y | N | Trunk and UE | | 75 | | 92/57, 79 |
| P34 | 1 | Y | N | Abdomen | 45 | | | 97/52, 108 |
| P34 | 3 | Y | N | Left trunk | | 160 | | 101/62, 99 |
| P1 | 1 | Y | N | Abdomen | 165 | 165 | 35 | 129/68, 69 |

*Y yes, N no, SBP systolic blood pressure, DBP diastolic blood pressure, HR heart rate, LE lower extremity, UE upper extremity.*

**Table 3 Incidence of risks associated with transcutaneous spinal cord stimulation.**

| Risks | # of instances (% occurrences) | Likelihood of event based on probability |
|---|---|---|
| Lower extremity motor responses | 14 (64%) | Likely to occur |
| Skin redness under stimulating electrodes | 9 (41%) | May occur half of the time |
| Pain from stimulation | 5 (22%) | Unlikely to occur |
| Autonomic dysreflexia | 1 (5%) | Very unlikely to occur |
| Numbness | 0 (0%) | Very unlikely to occur |

*% occurrence = # of instances/ # of assessments.*
*Total of 22 assessments in 8 children.*

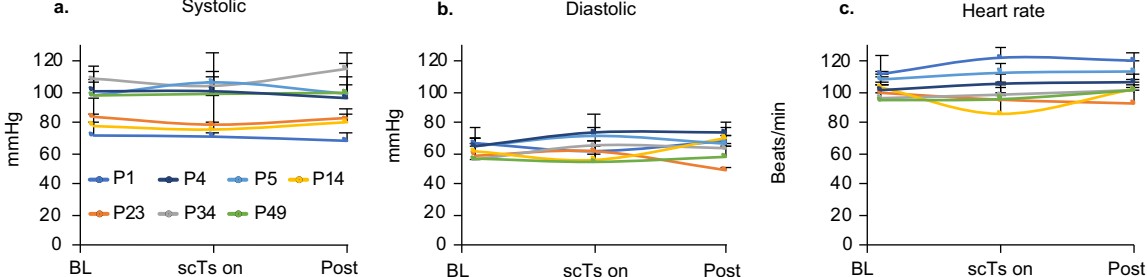

**Fig. 1 Hemodynamic parameters in response to acute transcutaneous spinal cord stimulation (scTs) in children with spinal cord injury (SCI).** Systolic (**a**) and diastolic (**b**) blood pressure (millimeter of Mercury, mmHg) and heart rate (**c**) (beats/minute) measurements during experiments at baseline (BL, 1 trial), with scTS at T11, L1 (and C5 when tolerated, 1 trial) and at the end of experiment (Post, 1 trial) graphed as means + standard deviation over 3 days of assessments for each participant ($n = 7$). No significant changes were observed in the systolic ($F_{2,12} = 1.98$, $p = 0.18$), diastolic ($F_{2,12} = 2.27$, $p = 0.1462$) blood pressures or heart rate ($F_{2,12} = 0.2$, $p = 0.82$) between the designated assessment time points at baseline, with scTS on and post experiment.

time points at baseline, with scTS on and post experiment in the participants (Fig. 1). Hip and/or knee extensor (10 instances) and flexor (4 instances) motor responses (Table 3) were observed in 7 children at the higher scTS intensities (130–170 mA). Unilateral or bilateral alternating ankle flexion/extension movements were also observed occasionally in some participants but were not counted as a risk since they were less vigorous and did not cause a loss of balance. Skin redness at the end of the experiment was observed in 9 out of 22 assessments (41%) which was dissipated by the next day according to a parental report on a follow-up inquiry.

**Acute effects of scTS on segmental trunk extension (proof-of-principle).** Transcutaneous spinal stimulation at both T11 and L1 significantly increased extension of the lower thoracic and lumbosacral trunk segments followed by an increase in flexion of the upper thoracic–cervical segments enabling overall neutral spine and an upright sitting posture in contrast to the baseline relaxed sitting posture (BL) immediately before stimulation or the participant's volitional attempt (VA) to sit upright without stimulation

(Figs. 2 and 3 [35]). Overall, there were significant timepoint differences (BL, VA, and scTs at T11) within the trunk kinematics and center of pressure (COP) displacement (interaction $F_{16,140} = 14.41$, $p < 0.0001$) (Fig. 3a). Specifically, during stimulation at T11, L5S1 was significantly more extended as compared to BL sitting ($p = 0.03$), PelvisT8 was significantly more extended during T11 scTS as compared to BL ($p < 0.0001$) and VA ($p < 0.0001$), and T8Head was significantly more extended during T11 scTS as compared to BL ($p = 0.0027$). T8Head was significantly more extended during VA ($p < 0.0001$) as compared to T11 scTS. In addition, the T8Head angle was significantly more extended during VA sitting as compared to BL ($p = 0.034$). There were no significant timepoint differences in other trunk segments. There was a significant change in anteroposterior COP from baseline during T11 scTS ($p = 0.0008$) (Fig. 3c).

Likewise, there were significant overall timepoint differences between BL, VA, and scTS at L1 in trunk kinematics and COP displacement (interaction $F_{16,140} = 7.75$, $p < 0.0001$) (Fig. 3b). Specifically, during stimulation at L1, L5S1 was significantly more extended as compared to BL sitting ($p = 0.006$) and VA

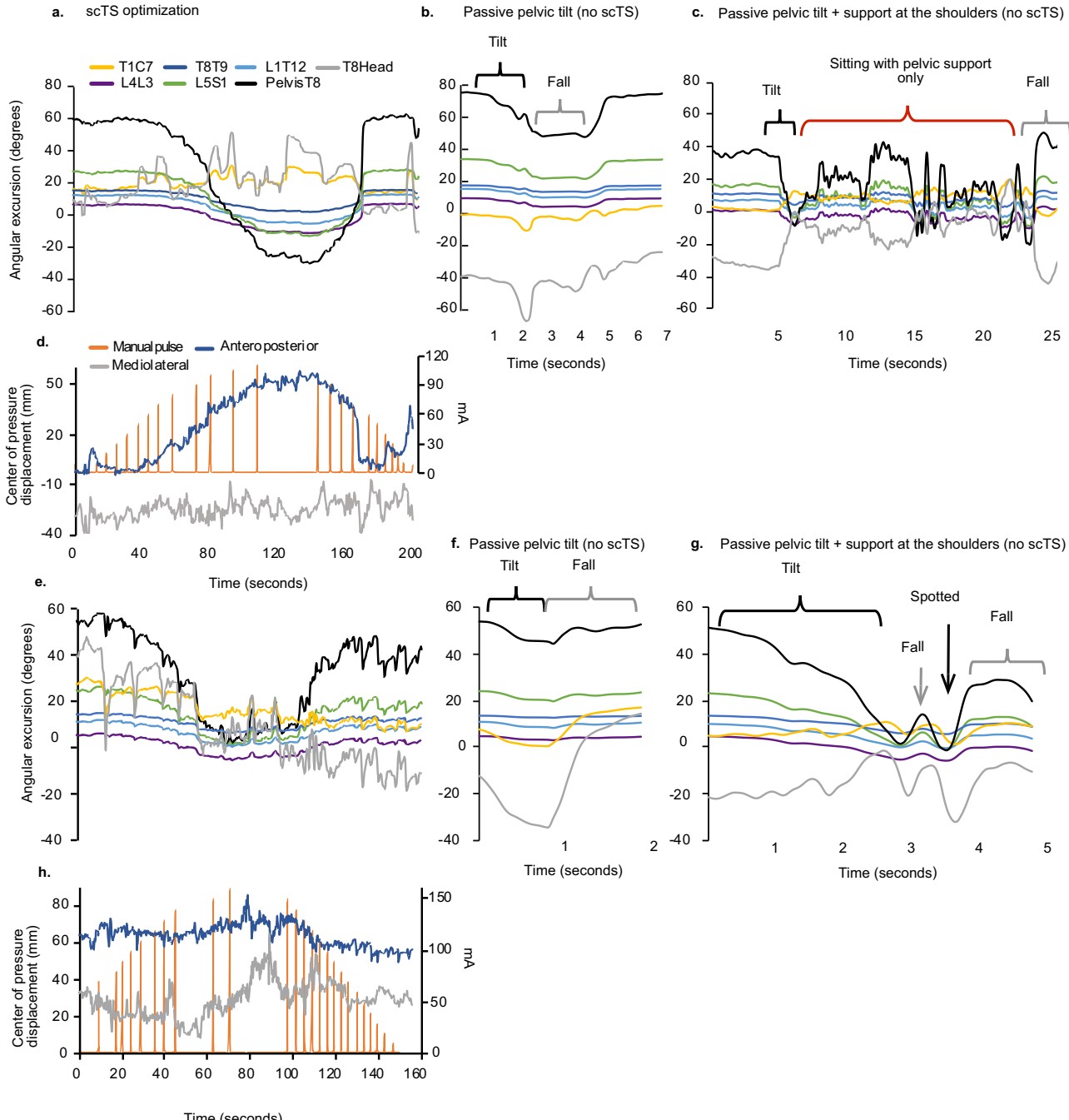

**Fig. 2 Sitting posture during transcutaneous spinal cord stimulation (scTS) vs. passive pelvic tilt in children with SCI.** Segmental trunk kinematics during scTS optimization for representative participants P14 (**a**) and P23 (**e**). Anteroposterior and mediolateral center of pressure displacements (millimeters, (mm)) were recorded concomitant with kinematics P14 (**d**), P23 (**h**). Manual pulse indicates the increase of stimulation intensity in 10 milliamp (mA) increments. Trunk kinematics during passive pelvic tilt was performed by a physical therapist without scTS while participants were seated relaxed, P14 (**b**) and P23 (**f**). Black curly bracket indicates the attempt to shift pelvis from posterior tilt toward neutral position, gray curly bracket indicates participants response of falling forward. Additional support at the anterior aspect of the shoulders was then provided by another therapist during passive pelvic tilt for P14 (**c**) and P23 (**g**). The participants were instructed to maintain upright posture after the shoulder support was removed with just the pelvic support. The red curly bracket indicates the participant's P14 attempt to stay upright (**c**). Participant P23 was not able to maintain balance once shoulder support was removed (**g**). Gray arrow points to the perturbation in trunk kinematics at the initial fall. The participant was spotted (black arrow) and repositioned to upright posture which P23 could not maintain, falling forward again when shoulder support was withdrawn.

($p = 0.03$), PelvisT8 was significantly more extended during L1 scTS as compared to BL ($p < 0.0001$) and VA ($p < 0.0001$), and T8Head was significantly more extended during L1 scTS as compared to BL ($p = 0.002$) and during VA as compared to BL

($p = 0.0002$). T8Head and T1C7 segments were significantly more extended during VA as compared to L1 scTS ($p < 0.0001$ and $p = 0.047$, respectively). There were no significant timepoint differences in other trunk segments. There was a significant

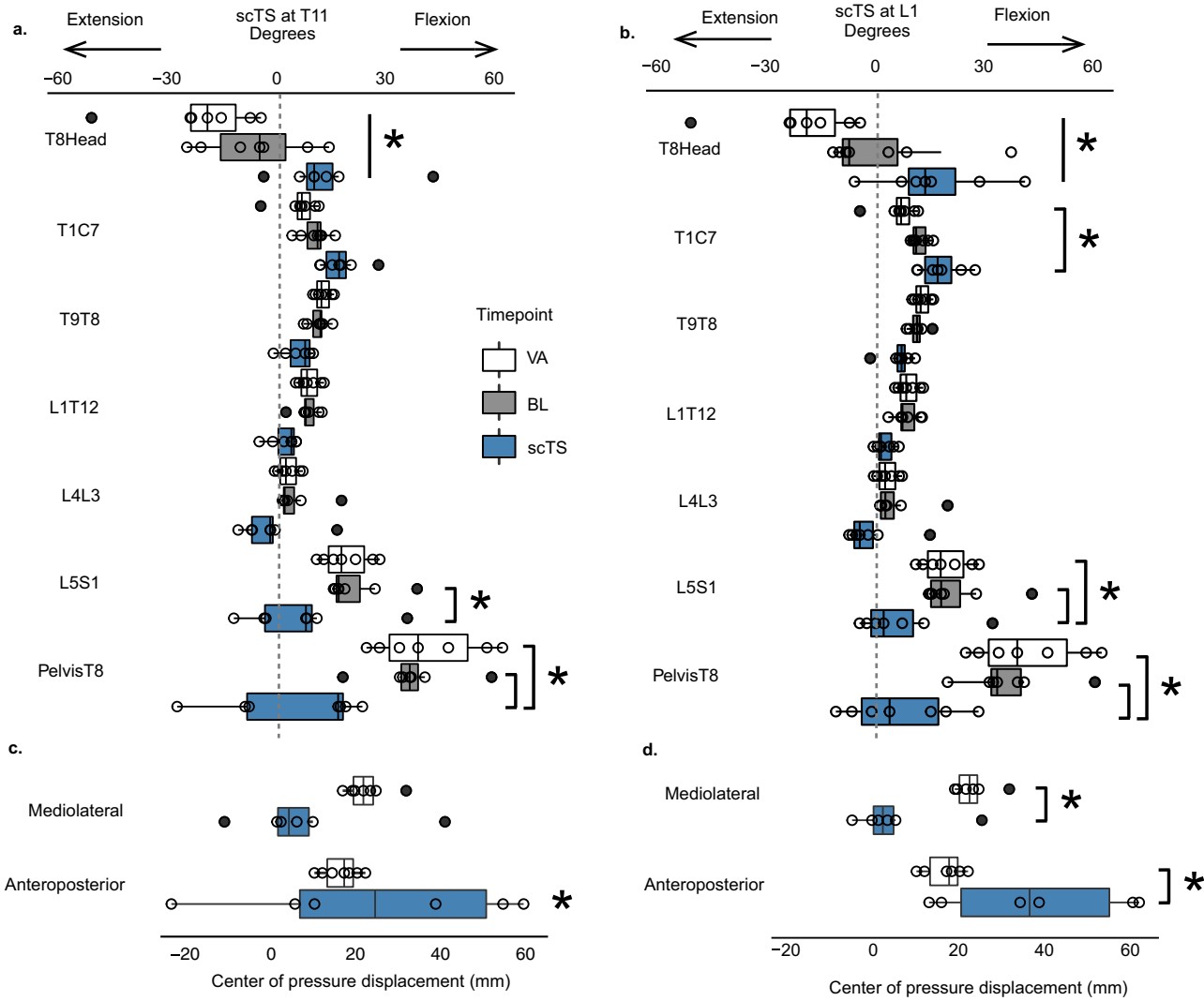

**Fig. 3 Acute effects of lumbosacral transcutaneous spinal cord stimulation on segmental trunk extension in children with SCI.** Box plots of angles at each measured trunk segment averaged over 10 s of the participants' volitional attempt (VA) to sit upright (white boxes) (average of 3 trials), prior to stimulation, baseline (BL) resting sitting (gray boxes) or during 10 s of sitting with scTS at T11 (**a**) and L1 (**b**) at the upright posture-inducing, scTS intensities optimized for each individual (blue boxes) ($n = 7$ participants for each stimulation site, 1 trial at each stimulation site on the third day). Change in anteroposterior and mediolateral center of pressure displacement (COP, millimeters, mm) during scTS at T11 (**c**) and L1 (**d**) relative to baseline ($n = 6$ for each stimulation site, the missing data point from one of the experiments occurred due to the loss of signal from the force plate). The centerline represents the group median, the left, and right box bounds represent 25th and 75th interquartile range (IQR), respectively. Box whiskers represent 1.5 times the IQR. The overlaying dots represent individual data points. Black dots are outlier points that lie outside of 1.5 times the IQR. The dotted line at 0° represents neutral vs. extended (−) or flexed (+) trunk position. Mixed linear regression models were used to assess the overall differences in trunk angles and COP changes across the timepoints of baseline sitting, volitional attempt, compared to stimulation at T11 ($F_{16,140} = 14.41$, $p < 0.0001$) and L1 ($F_{16,140} = 7.75$, $p < 0.0001$), followed by Tukey's post hoc $t$ test. *denotes significance for L5S1: BL vs. T11 scTS, $p = 0.03$, Cohen's $d = 0.97$; BL vs. L1 scTS, $p = 0.006$, Cohen's $d = 1.18$; VA vs. L1 scTS, $p = 0.03$, Cohen's $d = 0.97$; for PelvisT8: BL vs. T11 scTS, $p < 0.0001$, Cohen's $d = 2.1$; VA vs. T11 scTS, $p < 0.0001$, Cohen's $d = 2.4$; BL vs. L1 scTS, $p < 0.0001$, Cohen's $d = 2.35$; VA vs. L1 scTS, $p < 0.0001$, Cohen's $d = 2.76$; for T8Head: BL(T11) vs. VA, $p = 0.03$, Cohen's $d = 0.95$; BL vs. T11 scTS, $p = 0.0027$, Cohen's $d = 1.28$; VA vs. T11 scTS, $p < 0.0001$, Cohen's $d = 2.23$; BL (L1) vs. VA, $p = 0.0002$, Cohen's $d = 1.53$; BL vs. L1 scTS, $p < 0.002$, Cohen's $d = 1.33$; VA vs. L1 scTS, $p < 0.0001$, Cohen's $d = 2.86$; for anteroposterior COP: change from BL vs. T11 scTS, $p = 0.0008$, Cohen's $d = 1.53$, change from BL vs. L1 scTS, $p < 0.0001$, Cohen's $d = 2.95$; VA vs. L1 scTS, $p = 0.0001$, Cohen's $d = 1.74$; for mediolateral COP: VA vs. L1 scTS, $p = 0.0023$, Cohen's $d = 1.4$.

change in anteroposterior COP displacement from baseline to scTS ($p < 0.0001$) and during VA as compared to L1 scTS ($p = 0.0001$). There was also a significant change in mediolateral COP displacement during VA as compared to scTS ($p = 0.0023$) (Fig. 3d). We noted that stimulation at L1 was less effective at inducing trunk extension in a participant with a history of a 16° structural scoliosis and up to 57° positional curve. Stimulation at C5 at the tested intensities had no measurable effect on trunk extension or COP displacement in the participants who tolerated

cervical stimulation consistently across the three days (Supplementary Fig. 1).

**Effects of passive pelvic tilt (without scTS) on the capacity to sit upright.** A possible explanation for trunk extension in the lower lumbar and upper thoracic areas is a passive, biomechanical stacking of the vertebral segments initiated by the anterior tilting of the pelvis induced by local lumbosacral scTS. Thus, we tested

the effects of passive pelvic tilt on the upright posture in two of the study participants. First, passive pelvic tilt initiated while the child was sitting relaxed was a significant postural perturbation that did not lead to the upright vertebral column alignment, and at less than 20° from initial position resulted in a complete loss of balance and a fall forward in both of the tested participants. The same outcome was achieved even when the participants were encouraged to sit upright and try to maintain balance as their pelvis was passively tilted into a neutral position (Fig. 2b, f). Biomechanical stacking of the vertebral column was only possible if the participant was provided shoulder support at the anterior aspect at the same time as the other physical therapist performed passive pelvic tilt to prevent the forward fall. Participant P14 was able to achieve postural equilibrium with just pelvic support once the shoulder support was removed after the upright position was established. However, we noted that P14 had to use a compensatory abduction of his arms which he moved continuously during the trial in order to compensate for the lack of balance, as evident in turbulent trunk kinematics and the final collapse after effort-filled 25 s of sitting (Fig. 2c). Participant P23, who also has severe arm impairment, could not remain upright and as soon as the shoulder support was removed, the participant rapidly collapsed (Fig. 2g).

## Discussion

The current study investigated the safety and feasibility of scTS application in children with SCI. This was accomplished across 22 experiments in 8 children with SCI ages 3–14 years old.

One of the obvious concerns regarding the initial feasibility of neuromodulation using electrical spinal stimulation in children is whether discomfort or pain may limit the efficacy of this technique. Stimulation at T11 and L1 was well tolerated by 7 out of 8 children. Participant P21 with SCI at T11, perceived stimulation at T11 as painful even at very low intensities that previously in our study were not found to initiate any visible change in trunk extension. A diagnosis of allodynia provided an explanation for this participant's experience and identifies a potential risk factor for use of scTS in the population with SCI. The rest of the participants when asked described their perception of stimulation (if any) in the lumbosacral region as a vibration or tickling sensation. Most participants had increased sensitivity (subjective reports) in the region of cervical stimulation therefore the range of tested scTS intensities was relatively lower at C5 compared to those at T11 and L1. Cervical stimulation at the comfortable intensity level did not visibly produce a change in thoracic or lumbosacral extension (Supplementary Fig. 1). Overall, the participants in the current study across all ages had a positive response to scTS which is indicated by a 92% compliance rate.

With regards to safety, hemodynamic parameters were monitored throughout the study with the primary goal of tracking the incidence of autonomic dysreflexia (AD)[36,37]. We carefully monitored the participants for any typical signs of AD (e.g., sudden onset of facial flushing, headache)[38] throughout the experiments. One episode of AD occurred during scTS at the C5 in the youngest participant, age 3. Autonomic dysreflexia occurs in 51% of children with SCI and is most commonly associated with bowel impaction or bladder distention[38]. In general, any noxious sensory input arising below the level of injury may lead to the onset of AD[39]. Furthermore, innocuous sensory input may also trigger AD following SCI[40,41]. The evidence from basic animal studies suggests that maladaptive plasticity such as nociceptive afferent sprouting and other changes in spinal circuitry contribute to the autonomic dysregulation commonly seen in individuals with SCI[42,43]. Therefore, a child's hemodynamic stability should be monitored and ensured throughout the training

session regardless of the modes of therapy used in rehabilitation. Overall, stimulation in this study did not adversely affect hemodynamic parameters.

The most common observation during the increase in scTS intensity was robust hip and/or knee extension at higher stimulation intensities (64% of occurrences). This was not surprising as T11 and L1 stimulation electrodes deliver currents to the lumbosacral region that contains neural circuitry innervating lower extremities. Although these motor responses did not cause pain or participant discomfort, we report them as risks due to their sudden onset that could potentially cause a loss of balance in an individual with poor trunk control. Skin redness following stimulation was observed repeatedly in three out of eight participants for a total of eight instances during acute scTS application. In all cases, skin redness was not associated with any discomfort in that area and dissipated within 2 h or by the next day at parent follow-up. This observation is consistent with previous adult studies with the use of scTS[33,44].

As a proof-of-principle, we determined that scTS at either T11 (7 out of 7 participants) or L1 (6 out of 7 participants) produced an immediate change from a flexed or "C"-curve sitting posture to an upright posture at higher stimulation intensity similar to what has been observed in adults[30,45,46]. We noted that the stimulation intensities to achieve upright posture in our study were substantially higher than those previously reported in noninvasive spinal stimulation with traditional monophasic waveforms used to generate motor evoked potentials[47]. This is largely due to 10 kHz carrier frequency which splits 1 ms pulse duration into 10 biphasic pulses of 100 ms duration[48]. The shorter pulse durations are necessary to avoid nociceptive afferent activation to achieve pain-free stimulation[49,50]. The main parameter that determines stimulus strength and the subsequent motor fiber recruitment/torque generation is phase charge. Phase charge is a product of current amplitude and pulse duration[49,51–53]. To compensate for the reduced phase charge due to shorter pulse width with 10 kHz carrier frequency used in our stimulation paradigm, greater intensity of stimulation was required to generate motor responses[51,52,54,55].

We assessed changes in trunk kinematics and COP displacements in three conditions: during 10 s of a VA to sit upright, baseline relaxed sitting prior to stimulation, and sitting with scTS on at optimal stimulation intensity. The only differences in the trunk kinematics between VA and baseline sitting were reflected in the greater extension of the cervical region that corresponded to higher anteroposterior and mediolateral COP displacements during the VA. This finding suggests that the participants attempt to sit upright by hyperextending the neck. This compensatory strategy, however, does not carry over into the extension of the trunk segments below the level of injury, despite the participant's substantial effort, as reflected in changes in the COP displacement.

In the intact central nervous system, descending supraspinal pathways (e.g., vestibulospinal tract), provide tonic excitatory drive across the spinal neural axis where it is integrated with proprioceptive afferent input to achieve trunk stability and task-specific postural adaptations[56–59]. Loss of this tonic input following SCI leads to postural instability and lack of trunk control[60]. Lumbosacral epidural stimulation in adults with chronic motor complete SCI enables volitional control of isolated leg movements and overground locomotion[12,13]. Underlying physiological mechanisms for these observations remain under investigation. Electrical spinal stimulation, however, may augment the excitability of the otherwise dormant circuitry increasing the probability of motor output in response to supraspinal drive transmitted via spared descending axons[13,28]. Although the same physiological mechanism may underlie scTS-induced

upright posture, augmentation of supraspinal (corticospinal) drive via scTS is unlikely to be the primary explanation for our observation. To test the efficacy of scTS alone to improve sitting posture, the participants in the current study were instructed to sit relaxed and let the stimulation passively extend their trunk, minimizing the contribution of supraspinal influence. A major technical limitation of this study is the inability to determine the exact neural structures activated by the continuous electrical current. Previous neurophysiological investigations in adults have provided evidence that scTS evokes motor responses in the lower extremity muscles via polysynaptic projections of the dorsal root afferents that are directly activated by the scTS current, however, those studies employed single monophasic square wave pulses as opposed to the continuous high frequency modulated current used in our study[20,45,61–64]. To our knowledge, trunk muscle motor evoked potentials (MEP) thus far have been only studied using transcranial magnetic stimulation, as spinally evoked trunk muscle MEPs are masked by the spinal stimulation artifact[30]. For the same reason, although we recorded paraspinal muscle electromyography (EMG) in our participants (4 electrodes places bilaterally at T10 and L5 levels) during continuous scTS, the EMG signal was completely saturated by the high-frequency stimulation artifact due to the close proximity of the recording and the stimulating electrodes, limiting our ability to profile the EMG responses at different stimulation intensities which could have provided interesting mechanistic insights.

An alternative explanation for trunk extension in the lower lumbar and upper thoracic areas is a passive, stacking of the vertebral segments initiated by the anterior tilting of the pelvis during lumbosacral scTS. This may result due to biomechanical coupling between vertebral segments rather than scTS-induced neuromuscular activation of the postural control circuitry above the stimulation site. We assessed whether a passive pelvic tilt alone can induce upright posture by providing a more favorable biomechanical alignment resulting in the stacking of the lower and upper thoracic segments. We found that in the two tested participants, passive pelvic tilt without stimulation or the additional support at the shoulders was a significant perturbation with both participants falling forward before the pelvis reached neutral position. Although passive pelvic tilt was not systematically assessed in every participant in the current study, P14 and P23 represent the overall trunk function of the included participants based on the Segmental Assessment of Trunk Control (SATCo) scores (Table 1). Despite the mechanical coupling between the vertebral bodies via intervertebral discs and ligaments, the spinal column on its own (without neuromuscular activation) also has substantial flexibility. This flexibility and adaptability likely contribute to the 100% occurrence of neuromuscular scoliosis (curve > than 10°) development in children with SCI injured below the age of 10[5,7].

Although scTS induced trunk extension was ubiquitous across the participants, for the youngest participant, scTS at L1 also initiated hip extension pushing his entire trunk backward. Given his short trunk stature, the stimulating electrode likely overlaid more than one spinal cord level. Stimulation at L1, thus, had less spatial resolution allowing electrical currents to reach lower extremity motor pools. The initial scTS frequency parameters of 30 Hz at T11 and 15 Hz at L1 were chosen based on the established safe and efficacious frequencies in the adult population[28,30]. For younger and/or leaner participants, P23, L1 scTS frequency of 30 Hz instead of 15 Hz was optimal as it more consistently evoked a full "smooth" trunk extension. We also noted that scTS stimulation at L1 did not induce trunk extension in a participant with neuromuscular scoliosis in the thoracolumbar area (P4). As neuromuscular scoliosis is a typical secondary complication following pediatric-onset SCI, testing of

alternative electrode placement and/or a number of stimulating electrodes along the spine that could mitigate or reduce a scoliotic curve acutely could be a potential avenue for investigation. Testing alternative electrode placement was beyond the scope of the current study.

Based on the accumulated evidence, intensive task-specific physical rehabilitation is necessary to activate and recover neuromuscular capacity particularly below the level of injury[19,25,29,65]. Neuromodulation during training appears to be key in the facilitation and augmentation of use-dependent plasticity of neuronal networks[12,13,66,67]. Thus, future prospective studies will need to establish the long-term efficacy of adjunct therapies that combine task-specific training for trunk control with neuromodulation in children with SCI.

In summary, this study investigated the safety, feasibility, and proof-of-principle of transcutaneous spinal stimulation to acutely enable upright sitting posture in a sample of eight children (ages 3–14) with impaired trunk control due to chronic SCI. The stimulation was delivered using a custom-designed experimental device that has been proven safe in previous investigations in adults with SCI[68–70]. We conclude that, first, lumbosacral scTS is pain-free and well-tolerated in children with SCI whose injury level is at least two segments above the placement of the stimulating electrodes (T11 and L1). For participants with skin hypersensitivity, as may occur at or near the level of injury[71,72], scTS can be perceived as uncomfortable or painful at even low current intensities that likely do not reach the spinal cord. The use of scTS in children with such a condition should be evaluated case-by-case to assess the risk-benefit ratio. Alternative electrode placement to avoid evocation of pain while potentiating upright sitting posture should be considered and explored. Second, in general, continuous scTS (5–20 min) does not adversely affect hemodynamic parameters. However, children should be closely monitored for any signs or symptoms of AD during scTS. Third, scTS at higher intensities may cause sudden hip, knee, and/or ankle extension. Thus, we advise that a child is closely monitored and guarded, particularly during scTS testing at or above upright posture-inducing intensities. Fourth, skin redness under the stimulating electrodes may occur. In this study, skin redness was non-consequential as it dissipated without special measures within a few hours. Fifth, as a proof-of-principle, lumbosacral stimulation acutely evoked multi-segmental trunk extension in children with SCI. The degree of the response and/or stimulation intensity to achieve upright posture may vary depending on age, height, the amount of subcutaneous adipose tissue, and presence and/or degrees of neuromuscular scoliosis.

## Methods

*Demographics.* This study is a registered clinical trial (NCT03975634). The University of Louisville Institutional Review Board (IRB) approved this study (IRB protocol #19.0377). The study design and conduct complied with all relevant regulations regarding the use of human study participants and was conducted in accordance with the criteria set by the Declaration of Helsinki. The Human Locomotion Research Center's Volunteer Database (IRB protocol #06.0647) was used to identify potential research volunteers based on eligibility criteria. Informed consent and assent were signed by legal guardians of children and by children above 7 years of age, respectively. Eight children (3 females and 5 males, years range 3–14) with chronic, acquired upper motor neuron SCI, moderate to severe trunk control deficit as assessed by the segmental assessment of trunk control (SATCo, score < 20)[25,73] were recruited to participate in the current study (Table 1). Exclusion criteria included the use of botox within the past 3 months, current oral baclofen use,

musculoskeletal impairment limiting the range of motion, unhealed fracture or other medical complication limiting participation in the study, prior surgery for scoliosis, congenital SCI, and total ventilator dependence. The first and the last participant enrollments dates were 6/12/2019 and 01/16/2020, respectively.

*Experimental design.* This is a pilot non-randomized prospective experimental study using a within-subject design. Power calculation to estimate a sample size of $n = 8$ participants was based on the published study assessing the effects of transcutaneous spinal cord stimulation on trunk control in adults with SCI[30]. Rath et al. found that the center of pressure displacement during sitting without support changed from no-stim to stim conditions from $4.74 + 5.41$ mm to $1.36 + 0.98$ mm. Assuming a moderate pre-post correlation of 0.5, the no-stim to stim standard deviation corresponding to that change is 4.99 (using: $SD_{pre-post}$ = square root of $(SD^2_{pre} + SD^2_{post} - 2*pre-post\ correlation*SD_{pre}*SD_{post})$). A sample size of 8 provides 80% power to detect an effect size of 1.2 corresponds to a no-stim to stim change in COP displacement of 5.99 mm (classified as very large on the Cohen's scale[74] extended by Sawilowsky[75]) on a pre-post continuous measure using 2-sided paired *t*-test with a significance level of 0.05. Safety and feasibility of transcutaneous spinal stimulation (scTS) were established under the umbrella of proof-of-principle experiments that scTS acutely potentiates upright sitting posture in children with impaired trunk control due to SCI. Each participant was assessed 3 times on 3 separate days. On days 1 and 2 initial scTS optimizations and testing of upright posture-inducing scTS parameters were performed. On day 3, kinematics and center of pressure data were collected in addition to the safety-related outcome measures. BP, heart rate (HR) (using ABPM-05, Meditech, Budapest, Hungary, or manually) and pain (using FACES scale for children <8 years old and VAS pain scale for children ≥8)[76] were assessed at three standardized time points within the experiment: baseline, with stimulation on at three sites T11, L1, and C5, and at the end of the experiment. In the instances when a participant reported stimulation at C5 to be uncomfortable or painful, C5 stimulation was turned off and the BP measures were taken with stimulation on at just the two sites T11 and L1. Additional BP and HR measurements were taken if a participant exhibited signs of autonomic dysreflexia (AD) (e.g., sudden onset of facial flashing/redness and goosebumps). The occurrence of spasms or any motor responses during stimulation was documented. Throughout the assessments, before, during, and after stimulation, the comfort and status of the participants were carefully monitored. All events were documented and also followed over the next 24 h for a status update with the parent/caregiver when indicated, i.e., skin redness under an electrode at experiment completion. Compliance in attendance to all experiments was documented.

*Participant preparation.* At the beginning of each experiment, the participant's skin was examined for any redness or rash, particularly in the areas of electrode placement. For scTS, 2.5-cm round electrodes Axelgaard PALs Platinum were placed midline between (i) T10 and T11, (ii) T12 and L1, and (iii) C4 and C5 spinous processes, as cathodes and two $5.0 \times 8$ cm$^2$ rectangular electrodes placed symmetrically on the skin over the iliac crests as anodes. All electrodes were checked for any defects in the insulation layer before placement and were never reused. On day 3, for full-body 3D kinematics, MVN BIOMECH Awinda MTW2-3A7G6 sensors (Xsens Technologies B.V. Enschede, Netherlands)[77,78] were secured using a headband, Velcro straps or tape and placed on the following body segments: head, sternum, pelvis, on the upper and lower legs, upper arms, forearms,

hands, and feet following the user's manual instructions for specific landmarks.

*Transcutaneous spinal stimulation optimization for upright sitting posture.* Following preparation, the participants were seated on a force plate (Burtec, FP4060-NC-1000) with hips and knees positioned at 90° and feet non-weight bearing. Two research technicians, one in front and one at the back, guarded the participant at all times throughout the experiment. A proprietary 5-channel transcutaneous stimulator[68] was used to deliver biphasic rectangular waveform current with 1-ms pulse width and 15–30 Hz frequency with 10 kHz modulated carrier frequency. Stimulation frequencies were chosen based on previous studies using scTS: T11 at 30 Hz, L1 at 15 Hz, and C5 at 30 Hz[30,31,44]. The rationale for including cervical stimulation specifically, comes from previous studies in healthy adults that demonstrated cervical stimulation to potentiate motor output in the lower limbs likely via amplification of residual descending drive and/or descending propriospinal system[69,79]. Optimization of stimulation intensity was performed at three different spinal levels separately, starting at T11, L1, and lastly at C5[30]. This order of stimulation location testing was kept consistent throughout the experiments (non-randomized). Generally, we performed one trial of stimulation ramp up at each stimulation location once the optimal frequency was established. If 15 Hz stimulation at L1 led to greater activation of hip extensors before full trunk extension was achieved, the stimulation frequency was adjusted to 30 Hz and the stimulation was then replicated with the new frequency. The average angles during baseline sitting prior to stimulation with the most optimal parameters were then included in the group average computation.

The participant was instructed to sit relaxed (or as he/she would normally sit without support) and report any discomfort while stimulation intensity was increased in 5–10 mA increments until the threshold for induction of upright posture was reached. Stimulation optimization (from stimulation on to stimulation off) for each electrode location took anywhere from 3 to 5 min, depending on the maximum stimulation intensity needed for each of the participants to achieve the upright sitting posture and whether the adjustment of the parameters (e.g., stimulation frequency) was needed. For T11 and L1 stimulation sites, the upright posture-inducing threshold was determined as the intensity where there was a visible increase in thoracic and lumbar trunk extension and achievement of upright sitting posture (T11: 140 mA ± 23.1, L1: 134 mA ± 40). Due to increased sensitivity in the neck region, the intensity of stimulation at C5 was increased in smaller increments and only to the point of comfortable tolerance based on the participants' verbal feedback (38 mA ± 20). After upright-posture inducing thresholds were determined individually for each of the electrode locations, stimulation was turned on at all three locations. This was achieved by, first, turning on and ramping up the stimulation at T11, then L1, and then C5 if tolerated by the participant. Stimulation intensity at T11 and L1 was kept 10–20 mA lower than the maximum stimulation intensity tested during stimulation optimization for each location separately to avoid a possibility of lower extremity motor responses evocation. The duration of stimulation at all three or two channels was maintained for 2–3 min, while the BP and pain assessments were taken and then ramped down, one channel at a time. Altogether, each participant received 15–20 min of stimulation.

The effects of passive pelvic tilt (without scTS) on the capacity to sit upright and/or maintain upright posture was assessed in two children (P14 and P23) with a representative SATCo score of 11/20. During the SATCO a child is seated with hips and knees

each at 90° angles and feet resting on the floor with the pelvis neutrally aligned and held manually or with straps. Manual trunk support is provided horizontally on the lateral sides of the trunk and is segmentally shifted down from the highest level of support (starting at the clavicle) until a point of trunk instability above the support is reached. The score of 11/20 indicates that a child is able to maintain static, active, and dynamic (assessed as the ability to resist a perturbation) control above the segmental support over lower ribs but can no longer do so when the horizontal lateral support is shifted to below ribs[25,73]. During the passive pelvic tilt experiment, the participants were seated on the force plate with feet unsupported, as during stimulation trials. The physical therapist firmly placed their hands on the pelvis, thumbs at the sacrum, and fingers on the iliac spines with the participant sitting in their typical, relaxed posture, i.e., pelvis posteriorly tilted with a kyphotic back or rounded position. The physical therapist then slowly moved the pelvis into a neutral, upright position while the child was closely guarded due to the risk of falling during the passive tilt. The child was first instructed to sit relaxed, as was done for the stimulation trials. After unsuccessful attempts to sit upright during a passive pelvic tilt, assistance at the shoulders was provided in order to help the participant obtain an upright posture during a passive pelvic tilt from the posterior to the neutral position. Once the participant was upright, the participants were asked to maintain an upright posture for as long as possible without the shoulder support while the physical therapist continued to provide full pelvis support ensuring a neutral pelvis position during attempts at independent sitting.

*Data processing and analysis.* The kinematic data were sampled at 60 Hz and collected in MVN: 2019.2.1 software (XML format). Anteroposterior and mediolateral center of pressure displacements were acquired in NccHReflex (Labview, National instruments) in binary (.bin) format at a sampling rate of 2000 Hz. To synchronize kinematic and center of pressure data a trigger pulse was sent into NccHReflex. The.bin and Xsens files were then converted into text format files (.fns) and Comma Separated (.csv) files using a custom-written program in C-sharp (Data Processor 8.9, 2019). Another custom-written, C-sharp code (Mvnx2csv 2021.03.09) was then used to combine these two files by sampling up the Xsens data from 60 to 2000 Hz. Kinematics and force plate data were exported from the acquisition software as text files and imported into LabChart 8.1.3 (ADInstruments, USA) where the joint angles and center of pressure displacements were visualized, and average peaks and troughs in the angular excursions and center of pressure displacements were calculated for the 10 s of stable baseline sitting and sitting with transcutaneous spinal stimulation on using the Data Pad feature. The hemodynamic parameters (BP and HR) read-outs from the ambulatory BP monitor were manually recorded. The measurements were averaged across the 3 days at each time point within the experiment in Excel (Microsoft Office 365 ProPlus, Excel version 2002)

Safety and feasibility were determined by the frequency count of anticipated and unanticipated risks with associated percentage and likelihood (see Table 3). The likelihood was classified as very unlikely to occur (0–10%), unlikely to occur (11–40%), may occur about half of the time (41–60%), likely to occur (61–90%), and very likely to occur (91–100%)[80]. Statistical analysis was performed using mixed linear models. The effects of scTS on hemodynamic parameters were examined by comparing BP and HR measures at the three-time points within the experiment: baseline, with scTS turned on at T11 and L1 following optimization, and at the end of the experiment without scTS (seven children, three assessments per child) across 3 days. The analytical model regressed hemodynamic

measures on day (1–3), time point within the experiment, and their interaction and included a random intercept and random slopes for day and time point for each participant. To assess proof-of-principle, acute effects of scTS on sitting posture, mean flexion/extension angles (generated using a Kalman filter (Xsens Kalman Filter, XKF) from a 3D reconstruction of body segment position) of the head–T8, C7–T1, T8–T9, T12–L1, L3–L4, L5–S1, T8–Pelvis trunk segments during at least 10 s of stable baseline sitting (pre-scTS) were compared to the mean trunk segment angles during sitting with scTS at the upright posture-inducing threshold for each site of stimulation (T11 and L1) separately. The COP displacement in anteroposterior (A–P) and mediolateral (M–L) directions was quantified as a change from COP values acquired during 30 s of baseline sitting to the COP values during sitting with individualized optimal scTS intensity that induced upright posture. The analytical model to assess the immediate effect of stimulation on trunk extension consisted of mixed linear regression models for each stimulation site (T11 and L1) of trunk angle degrees for each trunk segment, change in COP displacement in anteroposterior and mediolateral directions, timepoint (volitional attempt (VA) to sit upright, sitting with no stimulation and sitting with stimulation), and their interaction. The significance of the timepoints effect within trunk segments was assessed by evaluating the type III Test for the interaction term (providing an F-statistic). Changes between time points, 2 by 2, within each stimulation site, trunk segment, and COP displacements were evaluated with post hoc t tests from linear contrasts built on the interaction term and adjusted for multiple comparisons using the Tukey's method. Outcome measures were summarized using least square mean and standard deviation. The latter was calculated by summing the within child standard deviation with the across children standard deviations. All tests were 2-sided with a significance level of 0.05. Statistical analyses were performed in SAS 9.4 (SAS 9.4M6., Cary, NC).

**Reporting summary**. Further information on research design is available in the Nature Research Reporting Summary linked to this article.

## Data availability
The hemodynamic outcome measures, the incidence of pain, skin redness, autonomic dysreflexia, and other safety-related outcome measures, as well as trunk kinematic and center of pressure displacement time-series data during transcutaneous electrical spinal cord stimulation in children with SCI data generated in this study have been deposited in the Open Data Commons for Spinal Cord Injury (ODC-SCI.org) database and released with the permanent digital object identifier (DOI) numbers as two datasets (DOI#1: https://doi.org/10.34945/F5HP4N[34], DOI#2: https://doi.org/10.34945/F5NC7X[35]) under a creative commons BY (CC-BY) 4.0 license. These data are publicly available to all registered ODC-SCI users. The access can be obtained by creating an account using the institutional email address at ODC-SCI.org. The raw participant demographics-related data are protected and are not available due to data privacy laws. The processed outcome measures data used for the generation of figures and tables in the manuscripts are available at ODC-SCI. The data generated in this study are provided as a Source Data file. The Human Locomotion Research Center's Volunteer Database is an IRB-approved volunteer database managed through a web-link with public access to register as a potential research candidate/volunteer (https://victoryoverparalysis.org/participate-in-research/). The database is an IRB-approved volunteer database: IRB 06.0647: Development of the KSCIRC Translational Research Database for Potential Research Volunteers. Access to use the database for recruitment of research subjects is not public and not available to the public, but only via institutional IRB approval requested by researchers. Source data are provided with this paper.

## Code availability
The code referenced in the paper and used for data processing and synchronization can be made available per request. Source data are provided with this paper.

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

## Acknowledgements
Research reported in this publication was supported by pilot funding from the National Institutes of Health National Center of Neuromodulation for Rehabilitation, NIH/NICHD Grant Number P2CHD086844 which was awarded to the Medical University of South Carolina. The sponsor involvement was limited to the financial support of the proposed experiments and had no impact on the study design, data collection, analysis, or paper preparation. The contents are solely the responsibility of the authors and do not necessarily represent the official views of the NIH or NICHD. The authors also thank Kosair Charities and the Todd Crawford Foundation for partial financial support of the current project. We would like to also thank Lisa Clayton, Laura Mendez, Nicholas Foster, Laura Leon Machado, Mackenzie Goode-Roberts, Thomas Machine, Seth Hagan, Sophie Humphrey, Elena Scheibler for their contributions to this study.

## Author contributions
Anastasia Keller: conceptualized study design, executed experimental stimulation protocol, directed data analysis, and interpretation, prepared the paper. Goutam Singh: conceptualized study design, executed trunk control assessments, contributed to data analysis interpretation, revised paper. Joel Sommerfeld: led data acquisition, contributed to biomechanical data analysis. Molly King: facilitated safety-related outcomes data acquisition and analysis. Parth Parikh: facilitated data acquisition and processing. Beatrice Ugiliweneza: contributed to study design and executed statistical data analysis. Jessica D'Amico: contributed to study design conceptualization, revised paper. Yury Gerasimenko: contributed to study design conceptualization, revised paper. Andrea L. Behrman: pediatric research program director, conceptualized study design, secured grant funding, oversaw study execution, revised paper.

## Competing interests
Dr. Yury Gerasimenko has a shareholder interest in NeuroRecovery Technologies and Cosyma. He holds certain inventorship rights on intellectual property licensed by the regents of the University of California to NeuroRecovery Technologies and its subsidiaries. Dr. Anastasia Keller, Dr. Goutam Singh, Joel Sommerfel, Molly King, Parth Parikh, Dr. Beatrice Ugiliweneza, Dr. Jessica D'Amico, and Dr. Andrea L. Behrman declares no competing interests.
