## [Peer Review File · Nature Communications]

Noninvasive spinal stimulation safely enables upright posture in children with spinal cord injuryREVIEWER COMMENTS

Reviewer #1 (Remarks to the Author):

This pilot clinical trial investigates the safety and feasibility of noninvasive spinal stimulation in children with spinal cord injury. Eight children attended 3 experimental sessions during which noninvasive spinal stimulation was applied using a custom-designed experimental device that produces stimulation waveforms with a 10kHz carrier frequency. Stimulation was applied at T10/T11, T12/L1 and C4/C5 spinal processes (cathodes) with anodes placed over the iliac crests. The authors reported pain in three participants, specifically at the C4/5 electrode site, which caused one participant to be withdrawn from the trial. They also report one episode of AD, and some further early signs of AD in other participants. Overall, the authors report that spinal stimulation is safe and feasible in children with SCI. The authors additionally report that spinal stimulation at T11 and L1 significantly increased trunk extension of the lower thoracic and lumbosacral segments.

Spinal stimulation following spinal cord injury is a rapidly growing field due to the remarkable results reported in early clinical trials in adults with clinically complete SCI. This study is an important contribution to the field, as spinal stimulation may have significant benefits for children, and the safety of this intervention in children living with spinal cord injury has not yet been reported to my knowledge. However, I have some major concerns about the manuscript in its current form.

The authors claim that the lumbosacral trunk extension, caused by the spinal stimulation, "spread" rostrally at higher stimulation intensities, which they describe as a "reversal of trunk muscle paralysis", and attribute to activation of an ascending "propriospinal neural network". While the spinal stimulation clearly caused trunk extension, there is no evidence of the rostral "spread" described by the authors in the results presented. No EMG data of trunk muscles has been provided, and the small kinematic changes noted in all segments rostral to L5 (Fig, 2B and 3A/B) provide insufficient evidence. Kinematic analysis assumes rigid bodies, which is not true when multiple segments at the spine are being analysed. Kinematic data during passive pelvic movement would be required to provide some evidence that these kinematic changes are due to muscle activation.

It is unsurprising that spinal stimulation caused lumbosacral extension, due to activation of trunk muscle motor neurons by T11/L1 electrodes, which has not been acknowledged by the authors. In order to activate spinal networks, the posterior roots (or ascending pathways) need to be activated by the stimulation, and the authors provided no evidence of this: did the authors carry out posterior root reflexes/paired pulses in order to prove activation of posterior roots (and therefore spinal networks), and to define stimulation parameters? If so, this data should be presented in the manuscript. If not, the authors should justify why these procedures were not carried out, and acknowledge this limitation.

Specific comments:

Title: "The feasibility and safety of noninvasive spinal stimulation in children with spinal cord injury" would better reflect the content of the manuscript.

Abstract:

Line 30: there is a typo "runk"

Lines 38-39: some results to support this statement should be presented in the abstract.

Introduction:

Line 65: The grammar in this sentence needs to be corrected.

Methods:

Line 90: define AB-LT.

More details are needed on the protocol during each experimental session.

Lines 103-104: How many trials of stimulation were carried out at each electrode location? Was

stimulation applied at all 3 electrode locations simultaneously? If so, provide further (technical) details on how the stimulator was set up to achieve this. Were locations tested in a random order? How long was stimulation applied for during each trial, and at what point during stimulation were outcome measures taken? What were the participants instructed to do during stimulation? The authors should also justify their stimulation parameters, acknowledging the effects of electrical stimulation applied with a 10kHz waveform. The currents reported (140mA) are substantially higher than the currents typically used in non-invasive spinal stimulation with traditional waveforms (root reflexes can typically be observed at approximately 40-50mA). It would be useful to understand the differences in sensory and motor threshold when a 10kHz carrier frequency is used.

Results:

Lines 177-178: How many participants were these effects observed in and on how many occasions? Was the stimulation stopped or did these effects resolve during stimulation? Were they associated with any particular electrode location/combination?

Discussion:

Lines: 244-250: This paragraph is not supported by the data presented and needs to be re-written or removed.

Lines 265 – 267: This sentence is not supported by the data. In particular, the authors cannot claim that the spinal stimulation caused a "reversal" of trunk muscle paralysis".

The authors should acknowledge that the trunk extension observed during spinal stimulation may be due to activation of motor neurons innervating trunk musculature.

Reviewer #2 (Remarks to the Author):

Thank you for the opportunity to review this well-written manuscript. The research described in the manuscript is novel and timely, and will be a valued contribution to the literature on pediatric SCI rehabilitation. The authors have clearly outlined the application of transcutaneous stimulation (including parameter settings) as well as the occurrence and mitigation of adverse effects; both of which will inform future research into the use of transcutaneous spinal stimulation in children with SCI.

I have a few questions that may increase clarity and the interpretation of the findings.

1) Metrics of sitting posture are compared between a baseline period and an experimental period where the stimulation is applied (children are instructed to "sit relaxed and let the stimulation passively extend their trunk"). It appears the baseline period involved relaxed sitting - is this correct? If yes, why compare metrics to passive sitting rather than a child's volitional attempt to sit with the trunk extended, as shown in Fig.2.A.a? Comparing to the volitional attempt would demonstrate the effect of the stimulation beyond what a child is actively able to contribute to the movement. I was also wondering how the inclusion criterion related to trunk control (i.e. SATCo score <15) may have influenced this decision. The authors indicate the a score <15 represents moderate to severe deficits in trunk control, but would some of the participants have been able to actively extend their trunk to some degree?

2. The rationale for including stimulation at C4/C5 is provided, but not until the Discussion. It would be helpful if that rationale was moved to the Introduction. Also, the role of C4/C5 stimulation in the effects reported in this manuscript is not clear. It sounds like a motor threshold could not be reached at this stimulation location for most children, is that correct? If yes, is this why stimulation at C4/C5 is not included in the 3 time points that were tested (i.e. "baseline, with scTS turned on at T11 and L1 following optimization, and at the end of the experiment without scTS"). It isn't clear to me whether C4/C5 stim was provided during the time that sitting metrics were calculated.

Minor comment: There is a typo in the first sentence of the Abstract (i.e. "runk").

Reviewer #3 (Remarks to the Author):

The authors investigated the feasibility and safety of transcutaneous spinal cord stimulation (tSCS) in children with spinal cord injury (SCI) in order to improve trunk stability. This pilot study is well designed, and data were presented accordingly. The authors concluded tSCS was well-tolerated in children with SCI whose injury is above the electrode placement and that continuous tSCS does not major adverse effect. Congratulations for that. However, minor comments should be addressed:

The introduction is very well-written and address properly the actual state of art of spinal cord stimulation.

Methods/ demographics: In the clinical trial protocol (NCT 03975634) the sample size estimated was of n= 10. However, in the present pilot study only showed data of 8 children with SCI. Please, clarify this discrepancy.

Line 90: Please explain acronyms AB-LT

Results: authors state that skin redness at the end of the experiments was observed in 8 out of 22 assessments (36%) Line 183-185. However, in table 2 the percentage of skin redness is of 23%. Please add further information.

Discussion:

Because of the fact that the intensity of the stimulation is fixed above the motor threshold, the motor response observed (trunk extension) is due to the stimulation of the afferent fibers (spinal mechanisms) or caused by a direct depolarization of the motor units in the muscle?

No additional comments. Congratulations again.

Response to Reviewers

NCOMMS-20-29124B entitled “Noninvasive spinal stimulation safely enables upright posture in children with spinal cord injury

Please note that the recommendations and comments from the reviewers are bolded and our responses are not bolded to differentiate the two from one another. We have also provided line numbers for changes in the manuscript for ease in locating them.

Reviewer #1 (Remarks to the Author):

This pilot clinical trial investigates the safety and feasibility of noninvasive spinal stimulation in children with spinal cord injury. Eight children attended 3 experimental sessions during which noninvasive spinal stimulation was applied using a custom-designed experimental device that produces stimulation waveforms with a 10kHz carrier frequency. Stimulation was applied at T10/T11, T12/L1 and C4/C5 spinal processes (cathodes) with anodes placed over the iliac crests. The authors reported pain in three participants, specifically at the C4/5 electrode site, which caused one participant to be withdrawn from the trial. They also report one episode of AD, and some further early signs of AD in other participants. Overall, the authors report that spinal stimulation is safe and feasible in children with SCI. The authors additionally report that spinal stimulation at T11 and L1 significantly increased trunk extension of the lower thoracic and lumbosacral segments.

Spinal stimulation following spinal cord injury is a rapidly growing field due to the remarkable results reported in early clinical trials in adults with clinically complete SCI. This study is an important contribution to the field, as spinal stimulation may have significant benefits for children, and the safety of this intervention in children living with spinal cord injury has not yet been reported to my knowledge. However, I have some major concerns about the manuscript in its current form.

The authors claim that the lumbosacral trunk extension, caused by the spinal stimulation, “spread” rostrally at higher stimulation intensities, which they describe as a “reversal of trunk muscle paralysis”, and attribute to activation of an ascending “propriospinal neural network”. While the spinal stimulation clearly caused trunk extension, there is no evidence of the rostral “spread” described by the authors in the results presented.

We thank the reviewer for the constructive criticism and have responded to the issue the reviewer rightfully raises with the interpretation of our results by removing this paragraph from the discussion.

No EMG data of trunk muscles has been provided, and the small kinematic changes noted in all segments rostral to L5 (Fig, 2B and 3A/B) provide insufficient evidence.

Trunk EMG data was collected as part of the study, however, could not be analyzed due to the significant stimulation artifact recorded by the EMG electrodes. The stimulation artifact could not be filtered out without losing most of the EMG signal which would have made the EMG data uninterpretable. We have added this to the discussion as a major technical limitation of our study.

Kinematic analysis assumes rigid bodies, which is not true when multiple segments at the spine are being analysed. Kinematic data during passive pelvic movement would be required to provide some evidence that these kinematic changes are due to muscle activation.

Two of the participants were available to return for the assessment of the effects of passive pelvic tilt on trunk kinematics, to address the Review's valid point. We have updated figure 2 to include the additional data and have addressed this point throughout the manuscript.

Methods: Lines 166- 184: "The effects of passive pelvic tilt (without scTS) on the capacity to sit upright and/or maintain upright posture was assessed in two children (P14 and P23) with a representative SATCO score of 11/20. During SATCO a child is seated with hips and knees each at 90-degree angles and feet resting on the floor with the pelvis neutrally aligned and held manually or with straps. Manual trunk support is provided horizontally on the lateral sides of the trunk and is segmentally shifted down from the highest level of support (starting at the clavicle) until a point of trunk instability is reached. The score of 11/20 if a child is able to maintain static, active and dynamic (assessed as ability to resist a perturbation) control with the segmental support over lower ribs but can no longer do so when the horizontal lateral support is shifted to below ribs. During passive pelvic tilt experiment, the participants were seated on the force plate with feet unsupported, as during stimulation trials. The physical therapist firmly placed their hands on the pelvis, thumbs at the sacrum and fingers on the iliac spines with the participant sitting in their typical, relaxed posture, i.e., pelvis posteriorly tilted with a kyphotic back or rounded position). The physical therapists then slowly moved the pelvis into a neutral, an upright position. The child was first instructed to sit relaxed, as was done for the stimulation trials. After unsuccessful attempts to sit upright during a passive pelvic tilt, assistance at the shoulders was provided in order to help the participant obtain an upright posture during passive pelvic tilt. Once the participant was upright, the participants were asked to maintain upright posture for as long as possible without the shoulder support while the physical therapist continued to provide full pelvis support ensuring neutral pelvis position during independent sitting."

Results: Lines 267-283: "Effects of passive pelvic tilt (without scTS) on the capacity to sit upright

A possible explanation for trunk extension in the lower lumbar and upper thoracic areas is a passive, biomechanical stacking of the vertebral segments initiated by the anterior tilting of the pelvis induced by local lumbosacral scTS. Thus, we tested the effects of passive pelvic tilt on the upright posture in two of the study participants. First, passive pelvic tilt initiated while the child was sitting relaxed was a significant postural perturbation that did not lead to the upright vertebral column alignment, and at less than 20 degrees from initial position resulted in a complete loss of balance and a fall forward in both of the tested participants. The same outcome was achieved even when the participants were encouraged to sit upright and try to maintain balance as their pelvis was passively tilted into neutral position (Fig. 3 B, F). Biomechanical stacking of the vertebral column was only possible if the participant was provided shoulder support at the anterior aspect at the same time as the other physical therapist performed passive pelvic tilt to prevent the forward fall. Participant P14 was able to achieve postural equilibrium with just pelvic support once the shoulder support was removed after upright position was established. However, we noted that P14 had to use a compensatory abduction of his arms which he moved continuously during the trial in order to compensate for the lack of balance, as evident in turbulent trunk kinematics and the final collapse after effort-filled 25 seconds of sitting (Fig 3. C). Participant P23, who also has severe arm impairment, could not

remain upright and as soon as the shoulder support was removed, the participant rapidly collapsed (Fig. 3G)”

Discussion: Lines 364-377: “An alternative explanation for trunk extension in the lower lumbar and upper thoracic areas is a passive, stacking of the vertebral segments initiated by the anterior tilting of the pelvis during lumbosacral scTS. This may result due to biomechanical coupling between vertebral segments rather than scTS-induced neuromuscular activation of the postural control circuitry above the stimulation site. We assessed whether a passive pelvic tilt alone can induce upright posture by providing a more favorable biomechanical alignment resulting in stacking of the lower and upper thoracic segments. We found that in the two tested participants, passive pelvic tilt without stimulation or the additional support at the shoulders was a significant perturbation with both participants falling forward before the pelvis reached neutral position. Although passive pelvic tilt was not systematically assessed in every participant in the current study, P14 and P23 represent the overall trunk function of the included participants based on the SATCO scores (Table 1). During SATCO a child is seated with hips and knees each at 90-degree angles and feet resting on the floor with the pelvis neutrally aligned and held manually or with straps. Manual trunk support is provided horizontally on the lateral sides of the trunk and is segmentally shifted down from the highest level of support (starting at the clavicle) until a point of trunk instability is reached. The average SATCO score 10.5/20 represents the participants’ capacity to maintain static, active and dynamic control with the segmental support over lower ribs but can no longer do so when the horizontal lateral support is shifted to below ribs (Butler et al. 2010, Argetsinger et al. 2019). Despite the mechanical coupling between the vertebral bodies, the spinal column on its own (without neuromuscular activation) also has substantial flexibility, which likely contributes to the staggering 100% occurrence of neuromuscular scoliosis (curve > than 10°) development in children with SCI injured below the age of 10”

It is unsurprising that spinal stimulation caused lumbosacral extension, due to activation of trunk muscle motor neurons by T11/L1 electrodes, which has not been acknowledged by the authors.

We fully recognize that the novelty of our study does not lie in the observation of trunk extension in response to stimulation but rather in the translation of this important neuromodulation technique to the pediatric SCI population. Studies must also be conducted in the pediatric population to advance care and to assess not only safety and feasibility, but also utility.

Thank you for bringing up this oversight. Activation of trunk muscle motoneurons is a possible mechanism for the induction of upright posture. We have added the sentences acknowledging the possibility of direct activation of trunk muscle motoneurons with scTS stimulation.

Lines: 351-354: “The induction of upright posture likely occurred as a result of transsynaptic activation of spinal networks through the dorsal roots, although we cannot rule out the direct activation of the motoneurons at the higher stimulation intensities as a contributing factor”

Although the stimulation sites are not located along the entire spinal extensor muscles from occiput below but are located at the very lowest thoracic and upper lumbar vertebrae. Extension of the spine occurred across spine segments as opposed to only the very low thoracic/upper lumbar areas.

In order to activate spinal networks, the posterior roots (or ascending pathways) need to be activated by the stimulation, and the authors provided no evidence of this: did the authors carry out posterior root reflexes/paired pulses in order to prove activation of posterior roots (and therefore spinal networks), and to define stimulation parameters? If so, this data should be presented in the manuscript. If not, the authors should justify why these procedures were not carried out, and acknowledge this limitation.

We have not performed posterior root reflex testing using paired pulse stimulation protocols in the current study for several reasons. First, spinally motor evoked potentials (MEP) would only be feasible to acquire from the lower extremities and not the trunk muscles due to the close proximity of the stimulating and the recording electrode. The short latency of the trunk muscle MEP gets lost in the stimulation artifact. Second, spinally evoked MEP require the delivery of current using monophasic paired pulses, as opposed to the biphasic, modulated continuous current we used in the current study. Doublet monophasic stimuli are not always well-tolerated in the adult sensate participants as they can be quite painful, and although most of our pediatric participant with SCI had no sensation at the lumbosacral level where the stimulation was delivered, we did not find it necessary at this stage to subject our participants to this protocol, given that there has been substantial evidence generated in the adult population demonstrating that scTS indeed causes activation of the spinal neural networks via dorsal root afferent depolarization (for cervical (arms) and lumbar motor pools (legs)). We have brought up these points in the discussion providing references for the latter (lines 354 – 363):

“A major technical limitation of this study is inability to determine the exact neural structures activated by the continuous electrical current. Previous neurophysiological investigations in adults have provided evidence that scTS evokes motor responses in the lower extremity muscles via polysynaptic projections of the dorsal root afferents that get directly activated by the scTS current. To our knowledge, trunk muscle motor evoked potentials (MEP) thus far have been only studied using transcranial magnetic stimulation, as spinally evoked trunk muscle MEPs are masked by the spinal stimulation artifact. For the same reason, although we recorded paraspinal muscle electromyography (EMG) in our participants (4 electrodes places bilaterally at T10 and L5 levels) during continuous scTS, the EMG signal was completely saturated by the high frequency stimulation artifact due to close proximity of the recording and the stimulating electrodes, limiting our ability to analyze the EMG data”

We do not exclude the possibility of performing these experiments in the pediatric population with SCI in the future clinical trials, particularly if these experiments become necessary in order to figure out specific stimulation parameters to improve efficacy of neuromodulation as a therapeutic tool. We believe that the research and clinical rehabilitation community may benefit from the findings of our study related to safety and feasibility of scTS implementation in children with SCI regardless of the exact mechanisms underlying scTS-induced upright sitting posture.

Specific comments

Title: “The feasibility and safety of noninvasive spinal stimulation in children with spinal cord injury” would better reflect the content of the manuscript.

Thank you for the suggestion, we have changed the title to “Noninvasive spinal stimulation safely enables upright posture in children with spinal cord injury”

Abstract:**Line 30: there is a typo “runk”**

Thank you, we corrected the typo.

Lines 38-39: some results to support this statement should be presented in the abstract.

We have revised the abstract according with CONSORT guidelines for pilot clinical trial reporting and have added brief summary of the results of the study to the abstract.

Introduction:**Line 65: The grammar in this sentence needs to be corrected.**

Thank you, we have fixed the grammar by splitting this long sentence into two separate sentences.

Methods:**Line 90: define AB-LT.**

We have defined AB-LT as activity-based locomotor training.

More details are needed on the protocol during each experimental session.

We have added further details about the protocol to the methods sub-section “Transcutaneous Spinal Stimulation Optimization for Upright Sitting Posture” (starting at line 131)

Lines 103-104: How many trials of stimulation were carried out at each electrode location?

Lines 142-145: “Generally, we performed one trial of stimulation ramp up at each stimulation location once the optimal frequency was established. If 15 Hz stimulation at L1 led to greater activation of hip extensors before full trunk extension was achieved, the stimulation frequency was adjusted to 30Hz and the stimulation was then replicated with the new frequency”

Was stimulation applied at all 3 electrode locations simultaneously? If so, provide further (technical) details on how the stimulator was set up to achieve this. Were locations tested in a random order?

Lines 140- 142: “Optimization of stimulation intensity was performed at three different spinal levels separately, starting at T11, L1 and lastly at C5. This order of stimulation location testing was kept consistent throughout the experiments (non-randomized)”

Lines 157– 162: “After upright-posture inducing threshold were determined individually for each of the electrode locations, stimulation was turned on at all three locations. This was achieved by, first, turning on and ramping up the stimulation at T11, then L1, and then C5 if tolerated by the participant. Stimulation intensity at T11 and L1 was kept 10-20 mA lower than the maximum stimulation intensity tested during stimulation optimization for each location separately to avoid a possibility of lower extremity motor responses evocation”

How long was stimulation applied for during each trial, and at what point during stimulation were outcome measures taken?

Lines 162 – 165: “The duration of stimulation at all three or two channels was maintained for 2-3 minutes, while the blood pressure and pain assessments were taken and then ramped down, one channel at a time. Altogether, each participant received 15-20 min of stimulation”

What were the participants instructed to do during stimulation?

The original version of the manuscript states the instructions given to the participants in Lines 129-131: “The participant was instructed to sit relaxed (or as he/she would normally sit without support) and report any discomfort while stimulation intensity was increased in 5-10 mA increments until the threshold for induction of upright posture was reached”. In the revised version this sentence is in Lines 148-150.

The authors should also justify their stimulation parameters, acknowledging the effects of electrical stimulation applied with a 10kHz waveform. The currents reported (140mA) are substantially higher than the currents typically used in non-invasive spinal stimulation with traditional waveforms (root reflexes can typically be observed at approximately 40-50mA). It would be useful to understand the differences in sensory and motor threshold when a 10kHz carrier frequency is used.

The explanation for the inverse relationship between modulated frequency vs. intensity with the main objective of pain-free stimulation as based on previous literature was added in the discussion section (Lines 322-331): “We noted that the stimulation intensities to achieve upright posture in our study were substantially higher than those previously reported in non-invasive spinal stimulation with traditional monophasic waveforms used to evoke motor evoked potentials. This is largely due to the 10 kHz carrier frequency which splits the 1 ms pulse duration into 10 biphasic pulses of 100 microseconds duration⁵⁴. The shorter pulse durations are necessary to avoid nociceptive afferent activation to achieve pain-free stimulation. The main parameter that determines stimulus strength and the subsequent motor fiber recruitment/torque generation is phase charge. Phase charge is a product of current amplitude and pulse duration. To compensate for the reduced phase charge due to shorter pulse width with 10 kHz carrier frequency used in our stimulation paradigm, greater intensity of stimulation was required to generate motor responses.”

Results:

Lines 177-178: How many participants were these effects observed in and on how many occasions? Was the stimulation stopped or did these effects resolve during stimulation? Were they associated with any particular electrode location/combo?

We have summarized our observations of goosebumps in the table and added it to the manuscript (Table 3)

Discussion:

Lines: 244-250: This paragraph is not supported by the data presented and needs to be re-written or removed.

Thank you, we have modified the paragraph to be more consistent with the evidence presented.

Lines 265 – 267: This sentence is not supported by the data. In particular, the authors cannot claim that the spinal stimulation caused a “reversal” of trunk muscle paralysis”.

Point well taken, thank you, we have removed this sentence from the discussion.

The authors should acknowledge that the trunk extension observed during spinal stimulation may be due to activation of motor neurons innervating trunk musculature.

Thank you, we have made this revision (Lines: 351-354) “The induction of upright posture likely occurred as a result of transsynaptic activation of spinal networks through the dorsal roots, although we cannot rule out the direct activation of the motoneurons at the higher stimulation intensities as a contributing factor”

Reviewer #2 (Remarks to the Author):

Thank you for the opportunity to review this well-written manuscript. The research described in the manuscript is novel and timely, and will be a valued contribution to the literature on pediatric SCI rehabilitation. The authors have clearly outlined the application of transcutaneous stimulation (including parameter settings) as well as the occurrence and mitigation of adverse effects; both of which will inform future research into the use of transcutaneous spinal stimulation in children with SCI.

I have a few questions that may increase clarity and the interpretation of the findings.

1) Metrics of sitting posture are compared between a baseline period and an experimental period where the stimulation is applied (children are instructed to "sit relaxed and let the stimulation passively extend their trunk"). It appears the baseline period involved relaxed sitting - is this correct?

That is correct, baseline was relaxed sitting.

If yes, why compare metrics to passive sitting rather than a child's volitional attempt to sit with the trunk extended, as shown in Fig.2.A.a? Comparing to the volitional attempt would demonstrate the effect of the stimulation beyond what a child is actively able to contribute to the movement.

Thank you for this suggestion. During our initial experiment we collected the kinematic and force plate data when the participants actively attempted to sit upright, but have not included it in the manuscript with the original submission. We have added this data to Fig. 3 and have reanalyzed the data to include 3 timepoint comparisons: volitional attempt, baseline relaxed sitting and sitting with scTS applied at an intensity that induces upright-sitting.

I was also wondering how the inclusion criterion related to trunk control (i.e. SATCo score <15) may have influenced this decision. The authors indicate the a score <15

represents moderate to severe deficits in trunk control, but would some of the participants have been able to actively extend their trunk to some degree?

The average SATCO score of the participants who tolerated stimulation was 10.5/20 (we have added individual SATCO scores in Table 1). The SATCO is scored with a child in sitting at 90-90 degrees at hips/pelvis with feet on the ground. Manual support is provided to achieve a neutrally-aligned pelvis, then sequential levels of trunk control are provided to test trunk control capacity above the level of support. If need be, additional support may be provided between the test level and the pelvic support due to a long trunk or instability. At each of 7 levels, three aspects of control are tested: 1) ability to hold a static posture (head still and positioned forward), 2) active posture (during head turning left and right), and 3) respond to perturbations (nudges anterior (manubrium), posterior (C7), lateral (Head of humerus) with quick displacement (if any) and rapid return to upright. Trunk control is scored on a scale from 0-20 with 20 being the highest score and indicates the ability to sit upright without any pelvic support or without any support above and demonstrates control for static, active, and reactive elements. The population tested ranged from 8/20 to 19/20. (Butler et al. 2010, Argetsinger et al. 2019). With the inclusion of the kinematics and COP data during volitional attempts to sit upright as well as passive pelvic tilt data from two participants (with the representative SATCO scores of 11) requested by another reviewer, we can confidently respond that our participants had very limited ability to sit upright on their own and also had great difficulty maintaining upright posture without both manual support at the pelvis and some support of the trunk above the pelvis. Please note we have revised the SATCO score inclusion criteria from SATCO <15, to SATCO < 19, as the score of 15 was a typo from a previous version of study design.

2. The rationale for including stimulation at C4/C5 is provided, but not until the Discussion. It would be helpful if that rationale was moved to the Introduction.

Thank you for the suggestion, we have moved the sentence that provides the rationale for including cervical stimulation in our study to the methods section where we describe the stimulation protocol we used (Lines: 137-140): "The rationale for including cervical stimulation specifically, comes from previous studies in healthy adults that demonstrated cervical stimulation to potentiate motor output in the lower limbs likely via amplification of residual descending drive and/or descending propriospinal system."

Also, the role of C4/C5 stimulation in the effects reported in this manuscript is not clear. It sounds like a motor threshold could not be reached at this stimulation location for most children, is that correct?

Yes, this is correct, we have pointed that out in the manuscript in Lines 264-266: "Stimulation at C5 at the tested intensities had no measurable effect on trunk extension or COP displacement in any of the participants either when tested individually or when it was turned on in addition to T11 and L1 stimulation (data not shown)"

If yes, is this why stimulation at C4/C5 is not included in the 3 time points that were tested (i.e. "baseline, with scTS turned on at T11 and L1 following optimization, and at the end of the experiment without scTS").

The blood pressure was taken with stimulation on at all three sites unless the participant reported discomfort or pain with cervical stimulation. We have added the details about the stimulation protocol which will hopefully clarify this point.

Lines 157-165: "After upright-posture inducing threshold were determined individually for each of the electrode locations, stimulation was turned on at all three locations. This was achieved by, first, turning on and ramping up the stimulation at T11, then L1, and then C5 if tolerated by the participant. Stimulation intensity at T11 and L1 was kept 10-20 mA lower than the maximum stimulation intensity tested during stimulation optimization for each location separately to avoid evocation of lower extremity motor responses. The duration of stimulation at all three or two channels was maintained for 2-3 minutes, while the blood pressure and pain assessments were taken and then ramped down, one channel at a time. Altogether, each participant received 15-20 min of stimulation"

It isn't clear to me whether C4/C5 stim was provided during the time that sitting metrics were calculated.

The trunk kinematic and center of pressure displacement data were collected for stimulation optimization at each electrode location. Since cervical stimulation did not result in trunk extension or induction of the upright posture at the comfortable for the participant stimulation intensities, we did not include these data in the manuscript. We have made a statement regarding this point in lines 264-266: "Stimulation at C5 at the tested intensities had no measurable effect on trunk extension or COP displacement in any of the participants either when tested individually or when it was turned on in addition to T11 and L1 stimulation (data not shown)"

Minor comment: There is a typo in the first sentence of the Abstract (i.e. "runk").

Thank you, we corrected the typo.

Reviewer #3 (Remarks to the Author):

The authors investigated the feasibility and safety of transcutaneous spinal cord stimulation (tSCS) in children with spinal cord injury (SCI) in order to improve trunk stability. This pilot study is well designed, and data were presented accordingly. The authors concluded tSCS was well-tolerated in children with SCI whose injury is above the electrode placement and that continuous tSCS does not major adverse effect. Congratulations for that. However, minor comments should be addressed: The introduction is very well-written and address properly the actual state of art of spinal cord stimulation.

Methods/ demographics: In the clinical trial protocol (NCT 03975634) the sample size estimated was of n= 10. However, in the present pilot study only showed data of 8 children with SCI. Please, clarify this discrepancy.

We did report potential enrollment of 8-10 patients in clinicaltrials.gov. We anticipated enrolling 1-2 participants for practice session to elucidate methodological aspects of the trial, the time needed to complete experiments, participant preparation, data acquisition etc. Our data analysis plan was developed based on an enrollment of 8 participants to the formal, cross-sectional study which are included in the manuscript.

Line 90: Please explain acronyms AB-LT

Thank you, we have defined AB-LT as activity-based locomotor training.

Results: authors state that skin redness at the end of the experiments was observed in 8 out of 22 assessments (36%) Line 183-185. However, in table 2 the percentage of skin redness is of 23%. Please add further information.

Out of 8 children, 5 children repeatedly experienced skin redness. In table 2 we erroneously calculated the incidence as the number of children who experienced it, as opposed to the total number of occurrences over the 22 experiments as we reported in the results. In addition, we have gone back and checked our records and have identified 9 instead of 8 (as reported in the results of the original manuscript) total occurrences of skin redness in 5 children. Therefore, we have fixed both table 2 and the results section to reflect these revisions. We thank the reviewer for bringing this to our attention.

Discussion:

Because of the fact that the intensity of the stimulation is fixed above the motor threshold, the motor response observed (trunk extension) is due to the stimulation of the afferent fibers (spinal mechanisms) or caused by a direct depolarization of the motor units in the muscle?

Thank you for bringing up this point. Although we have not performed spinally evoked motor potentials in our studies to determine which neural structures are being activated by the scTS specifically, for a number of reasons, we believe that the scTS-induced trunk extension is primarily due to the activation of spinal networks via dorsal root afferents which are directly activated by the current. We also do not exclude the possibility of a direct motor neuronal activation at the higher stimulation intensities. We have added these points to the discussion, lines 351-354: "The induction of upright posture likely occurred as a result of transsynaptic activation of spinal networks through the dorsal roots, although we cannot rule out the direct activation of the motoneurons at the higher stimulation intensities as a contributing factor"

No additional comments. Congratulations again.

REVIEWERS' COMMENTS

Reviewer #1 (Remarks to the Author):

The authors have provided adequate replies to my comments, thank you. However, one point needs further clarification, regarding the structures being activated. In lines 355-358, the authors state: "Previous neurophysiological investigations in adults have provide evidence that scTS evokes motor responses in the lower extremity muscles via polysynaptic projections of the dorsal root afferents that are directly activated by the scTS current (20,53,69-72)". The motor responses in all of these studies were elicited with single monophasic, square-wave pulses. They did not incorporate the modulated 10kHz carrier frequency in these tests. In our own lab, we have attempted lower limb motor responses using single pulses with the modulated 10kHz carrier (in healthy participants), and the responses were elicited at much higher current intensities (~190mA in some participants, in others we could not elicit responses at <200mA, which was our upper current limit). In my opinion, the authors should acknowledge that the structures being activated with the stimulation parameters they employ may be different, and that they cannot determine from their data whether the stimulation being applied was above or below motor threshold for the dorsal roots. Given that the subjects did not use volitional effort to contribute to the upright posture, it seems most likely that the results were due to direct activation of motor neurons in the back musculature, rather than due to sub-threshold stimulation of dorsal roots.

I therefore recommend that the authors modify lines 351-354 to: "The induction of upright posture may have occurred as a result of transsynaptic activation of spinal networks through the dorsal roots and/or direct activation of the motoneurons."

Reviewer #2 (Remarks to the Author):

The authors have thoroughly addressed the reviewers' comments - thank you!

Reviewer #3 (Remarks to the Author):

Authors addressed nicely to the concerns raised by my review of the original manuscript. No additional comments.

REVIEWERS' COMMENTS

Reviewer #1 (Remarks to the Author):

The authors have provided adequate replies to my comments, thank you. However, one point needs further clarification, regarding the structures being activated. In lines 355-358, the authors state: "Previous neurophysiological investigations in adults have provide evidence that scTS evokes motor responses in the lower extremity muscles via polysynaptic projections of the dorsal root afferents that are directly activated by the scTS current (20,53,69-72)". The motor responses in all of these studies were elicited with single monophasic, square-wave pulses. They did not incorporate the modulated 10kHz carrier frequency in these tests. In our own lab, we have attempted lower limb motor responses using single pulses with the modulated 10kHz carrier (in healthy participants), and the responses were elicited at much higher current intensities (~190mA in some participants, in others we could not elicit responses at <200mA, which was our upper current limit). In my opinion, the authors should acknowledge that the structures being activated with the stimulation parameters they employ may be different, and that they cannot determine from their data whether the stimulation being applied was above or below motor threshold for the dorsal roots. Given that the subjects did not use volitional effort to contribute to the upright posture, it seems most likely that the results were due to direct activation of motor neurons in the back musculature, rather than due to sub-threshold stimulation of dorsal roots.

I therefore recommend that the authors modify lines 351-354 to: "The induction of upright posture may have occurred as a result of transsynaptic activation of spinal networks through the dorsal roots and/or direct activation of the motoneurons."

We thank the reviewer for their careful review of our work once again. We appreciate the points brought up in regard to deciphering the precise neural mechanisms underlying transcutaneous spinal cord stimulation, particularly those in reference to the effects of different wave forms of stimulation (continuous vs monophasic single pulse) on the recruitment of dorsal root afferents vs. direct motoneuronal activation. To resolve this valid concern, we have removed the sentence previously in lines 351-354: "The induction of upright posture may have occurred as a result of transsynaptic activation of spinal networks through the dorsal roots and/or direct activation of the motoneurons" from the manuscript.

In addition, we have pointed out that studies on motor evoked potentials using spinal cord stimulation in adults which we have referenced utilized a different current type (lines 367-368): "Previous neurophysiological investigations in adults provide evidence that scTS evokes motor responses in the lower extremity muscles via polysynaptic projections of the dorsal root afferents that are directly activated by the scTS current, however, those studies employed single monophasic square wave pulses as opposed to the continuous high frequency modulated current used in our study".

We hope that these edits, along with our previous revision that explicitly stated the limitation of our study (lines: 363-365: "A major technical limitation of this study is the inability to determine the exact neural structures activated by the continuous electrical current") will resolve the concern brought about by our speculations about the mechanisms.

We express once again the appreciation for the reviewer's valuable feedback and constructive criticisms of our manuscript. We acknowledge that the challenges posed through the review process have substantially improved our manuscript. Thank you.

Reviewer #2 (Remarks to the Author):

The authors have thoroughly addressed the reviewers' comments - thank you!

We thank the reviewer very much for their time and the feedback provided during the manuscript revision process. We acknowledge that the reviewer's effort has led to major improvements in our revised manuscript. Thank you.

Reviewer #3 (Remarks to the Author):

Authors addressed nicely to the concerns raised by my review of the original manuscript. No additional comments.

We thank the reviewer for the time and effort spent on the revisions of our manuscript and the valuable feedback that, we believe, have significantly improved our manuscript as a result. Thank you.